# Imagination-Augmented Agents
# for Deep Reinforcement Learning

**Sébastien Racanière**[*]   **Théophane Weber**[*]   **David P. Reichert**[*]   **Lars Buesing**
**Arthur Guez**   **Danilo Rezende**   **Adria Puigdomènech Badia**   **Oriol Vinyals**
**Nicolas Heess**   **Yujia Li**   **Razvan Pascanu**   **Peter Battaglia**
**Demis Hassabis**   **David Silver**   **Daan Wierstra**
DeepMind

## Abstract

We introduce Imagination-Augmented Agents (I2As), a novel architecture for deep reinforcement learning combining model-free and model-based aspects. In contrast to most existing model-based reinforcement learning and planning methods, which prescribe how a model should be used to arrive at a policy, I2As learn to interpret predictions from a learned environment model to construct implicit plans in arbitrary ways, by using the predictions as additional context in deep policy networks. I2As show improved data efficiency, performance, and robustness to model misspecification compared to several baselines.

## 1   Introduction

A hallmark of an intelligent agent is its ability to rapidly adapt to new circumstances and "achieve goals in a wide range of environments" [1]. Progress has been made in developing capable agents for numerous domains using deep neural networks in conjunction with *model-free* reinforcement learning (RL) [2–4], where raw observations directly map to values or actions. However, this approach usually requires large amounts of training data and the resulting policies do not readily generalize to novel tasks in the same environment, as it lacks the behavioral flexibility constitutive of general intelligence.

*Model-based* RL aims to address these shortcomings by endowing agents with a model of the world, synthesized from past experience. By using an internal model to reason about the future, here also referred to as *imagining*, the agent can seek positive outcomes while avoiding the adverse consequences of trial-and-error in the real environment – including making irreversible, poor decisions. Even if the model needs to be learned first, it can enable better generalization across states, remain valid across tasks in the same environment, and exploit additional unsupervised learning signals, thus ultimately leading to greater data efficiency. Another appeal of model-based methods is their ability to scale performance with more computation by increasing the amount of internal simulation.

The neural basis for imagination, model-based reasoning and decision making has generated a lot of interest in neuroscience [5–7]; at the cognitive level, model learning and mental simulation have been hypothesized and demonstrated in animal and human learning [8–11]. Its successful deployment in artificial model-based agents however has hitherto been limited to settings where an exact transition model is available [12] or in domains where models are easy to learn – e.g. symbolic environments or low-dimensional systems [13–16]. In complex domains for which a simulator is not available to the agent, recent successes are dominated by model-free methods [2, 17]. In such domains, the performance of model-based agents employing standard planning methods usually suffers from model errors resulting from function approximation [18, 19]. These errors compound during planning, causing over-optimism and poor agent performance. There are currently no planning

---

[*]Equal contribution, corresponding authors: {sracaniere, theophane, reichert}@google.com.

or model-based methods that are robust against model imperfections which are inevitable in complex domains, thereby preventing them from matching the success of their model-free counterparts.

We seek to address this shortcoming by proposing Imagination-Augmented Agents, which use approximate environment models by "learning to interpret" their imperfect predictions. Our algorithm can be trained directly on low-level observations with little domain knowledge, similarly to recent model-free successes. Without making any assumptions about the structure of the environment model and its possible imperfections, our approach learns in an end-to-end way to extract useful knowledge gathered from model simulations – in particular not relying exclusively on simulated returns. This allows the agent to benefit from model-based imagination without the pitfalls of conventional model-based planning. We demonstrate that our approach performs better than model-free baselines in various domains including Sokoban. It achieves better performance with less data, even with imperfect models, a significant step towards delivering the promises of model-based RL.

## 2 The I2A architecture

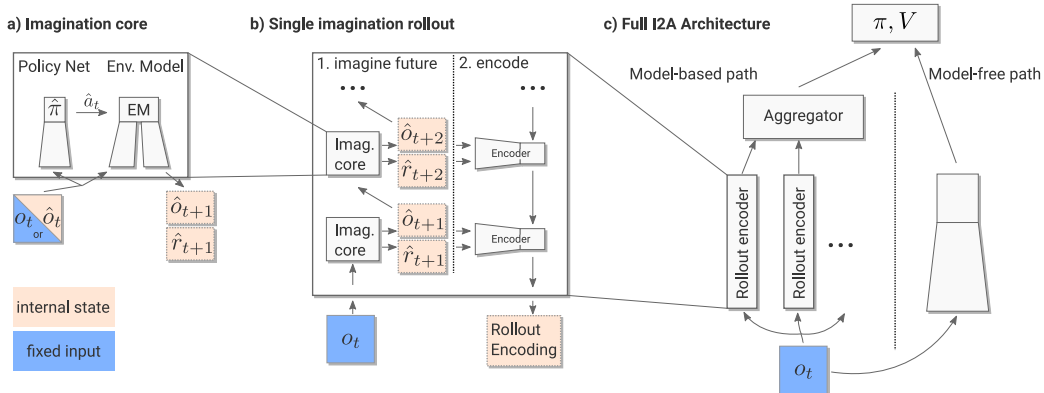

Figure 1: *I2A architecture*. $\hat{\cdot}$ notation indicates imagined quantities. *a)*: the imagination core (IC) predicts the next time step conditioned on an action sampled from the rollout policy $\hat{\pi}$. *b)*: the IC imagines trajectories of features $\hat{f} = (\hat{o}, \hat{r})$, encoded by the rollout encoder. *c)*: in the full I2A, aggregated rollout encodings and input from a model-free path determine the output policy $\pi$.

In order to augment model-free agents with imagination, we rely on environment models – models that, given information from the present, can be queried to make predictions about the future. We use these environment models to simulate *imagined trajectories*, which are interpreted by a neural network and provided as additional context to a policy network.

In general, an environment model is any recurrent architecture which can be trained in an unsupervised fashion from agent trajectories: given a past state and current action, the environment model predicts the next state and any number of signals from the environment. In this work, we will consider in particular environment models that build on recent successes of action-conditional next-step predictors [20–22], which receive as input the current observation (or history of observations) and current action, and predict the next observation, and potentially the next reward. We roll out the environment model over multiple time steps into the future, by initializing the imagined trajectory with the present time real observation, and subsequently feeding simulated observations into the model.

The actions chosen in each rollout result from a rollout policy $\hat{\pi}$ (explained in Section 3.1). The environment model together with $\hat{\pi}$ constitute the imagination core module, which predicts next time steps (Fig 1a). The imagination core is used to produce $n$ trajectories $\hat{\mathcal{T}}_1, \dots, \hat{\mathcal{T}}_n$. Each imagined trajectory $\hat{\mathcal{T}}$ is a sequence of features $(\hat{f}_{t+1}, \dots, \hat{f}_{t+\tau})$, where $t$ is the current time, $\tau$ the length of the rollout, and $\hat{f}_{t+i}$ the output of the environment model (i.e. the predicted observation and/or reward).

Despite recent progress in training better environment models, a key issue addressed by I2As is that a learned model cannot be assumed to be perfect; it might sometimes make erroneous or nonsensical predictions. We therefore do not want to rely solely on predicted rewards (or values predicted

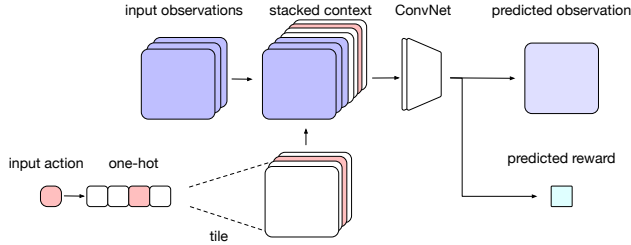

input observations · stacked context · ConvNet · predicted observation

input action · one-hot

tile

predicted reward

Figure 2: *Environment model.* The input action is broadcast and concatenated to the observation. A convolutional network transforms this into a pixel-wise probability distribution for the output image, and a distribution for the reward.

from predicted states), as is often done in classical planning. Additionally, trajectories may contain information beyond the reward sequence (a trajectory could contain an informative subsequence – for instance solving a subproblem – which did not result in higher reward). For these reasons, we use a rollout encoder $\mathcal{E}$ that processes the imagined rollout as a whole and *learns to interpret it*, i.e. by extracting any information useful for the agent's decision, or even ignoring it when necessary (Fig 1b). Each trajectory is encoded separately as a rollout embedding $e_i = \mathcal{E}(\hat{\mathcal{T}}_i)$. Finally, an aggregator $\mathcal{A}$ converts the different rollout embeddings into a single imagination code $c_{\text{ia}} = \mathcal{A}(e_1, \ldots, e_n)$.

The final component of the I2A is the policy module, which is a network that takes the information $c_{ia}$ from model-based predictions, as well as the output $c_{\text{mf}}$ of a model-free path (a network which only takes the real observation as input; see Fig 1c, right), and outputs the imagination-augmented policy vector $\pi$ and estimated value $V$. The I2A therefore learns to combine information from its model-free and imagination-augmented paths; note that without the model-based path, I2As reduce to a standard model-free network [3]. I2As can thus be thought of as augmenting model-free agents by providing additional information from model-based planning, and as having strictly more expressive power than the underlying model-free agent.

## 3 Architectural choices and experimental setup

### 3.1 Rollout strategy

For our experiments, we perform one rollout for each possible action in the environment. The first action in the i[th] rollout is the i[th] action of the action set $\mathcal{A}$, and subsequent actions for all rollouts are produced by a shared rollout policy $\hat{\pi}$. We investigated several types of rollout policies (random, pretrained) and found that a particularly efficient strategy was to distill the imagination-augmented policy into a model-free policy. This distillation strategy consists in creating a small model-free network $\hat{\pi}(o_t)$, and adding to the total loss a cross entropy auxiliary loss between the imagination-augmented policy $\pi(o_t)$ as computed on the current observation, and the policy $\hat{\pi}(o_t)$ as computed on the same observation. By imitating the imagination-augmented policy, the internal rollouts will be similar to the trajectories of the agent in the real environment; this also ensures that the rollout corresponds to trajectories with high reward. At the same time, the imperfect approximation results in a rollout policy with higher entropy, potentially striking a balance between exploration and exploitation.

### 3.2 I2A components and environment models

In our experiments, the encoder is an LSTM with convolutional encoder which sequentially processes a trajectory $\mathcal{T}$. The features $\hat{f}_t$ are fed to the LSTM in reverse order, from $\hat{f}_{t+\tau}$ to $\hat{f}_{t+1}$, to mimic Bellman type backup operations.[2] The aggregator simply concatenates the summaries. For the model-free path of the I2A, we chose a standard network of convolutional layers plus one fully connected one [e.g. 3]. We also use this architecture on its own as a baseline agent.

Our environment model (Fig. 2) defines a distribution which is optimized by using a negative log-likelihood loss $l_{\text{model}}$. We can either pretrain the environment model before embedding it (with frozen weights) within the I2A architecture, or jointly train it with the agent by adding $l_{\text{model}}$ to the total loss as an auxiliary loss. In practice we found that pre-training the environment model led to faster runtime of the I2A architecture, so we adopted this strategy.

For all environments, training data for our environment model was generated from trajectories of a partially trained standard model-free agent (defined below). We use partially pre-trained agents because random agents see few rewards in some of our domains. However, this means we have to account for the budget (in terms of real environment steps) required to pretrain the data-generating agent, as well as to then generate the data. In the experiments, we address this concern in two ways: by explicitly accounting for the number of steps used in pretraining (for Sokoban), or by demonstrating how the same pretrained model can be reused for many tasks (for MiniPacman).

### 3.3 Agent training and baseline agents

Using a fixed pretrained environment model, we trained the remaining I2A parameters with asynchronous advantage actor-critic (A3C) [3]. We added an entropy regularizer on the policy $\pi$ to encourage exploration and the auxiliary loss to distill $\pi$ into the rollout policy $\hat{\pi}$ as explained above. We distributed asynchronous training over 32 to 64 workers; we used the RMSprop optimizer [23]. We report results after an initial round of hyperparameter exploration (details in Appendix A). Learning curves are averaged over the top three agents unless noted otherwise.

A separate hyperparameter search was carried out for each agent architecture in order to ensure optimal performance. In addition to the I2A, we ran the following baseline agents (see Appendix B for architecture details for all agents).

**Standard model-free agent.** For our main baseline agent, we chose a model-free standard architecture similar to [3], consisting of convolutional layers (2 for MiniPacman, and 3 for Sokoban) followed by a fully connected layer. The final layer, again fully connected, outputs the policy logits and the value function. For Sokoban, we also tested a 'large' standard architecture, where we double the number of all feature maps (for convolutional layers) and hidden units (for fully connected layers). The resulting architecture has a slightly larger number of parameters than I2A.

**Copy-model agent.** Aside from having an internal environment model, the I2A architecture is very different from the one of the standard agent. To verify that the information contained in the environment model rollouts contributed to an increase in performance, we implemented a baseline where we replaced the environment model in the I2A with a 'copy' model that simply returns the input observation. Lacking a model, this agent does not use imagination, but uses the same architecture, has the same number of learnable parameters (the environment model is kept constant in the I2A), and benefits from the same amount of computation (which in both cases increases linearly with the length of the rollouts). This model effectively corresponds to an architecture where policy logits and value are the final output of an LSTM network with skip connections.

## 4 Sokoban experiments

We now demonstrate the performance of I2A over baselines in a puzzle environment, Sokoban. We address the issue of dealing with imperfect models, highlighting the strengths of our approach over planning baselines. We also analyze the importance of the various components of the I2A.

Sokoban is a classic planning problem, where the agent has to push a number of boxes onto given target locations. Because boxes can only be pushed (as opposed to pulled), many moves are irreversible, and mistakes can render the puzzle unsolvable. A human player is thus forced to plan moves ahead of time. We expect that artificial agents will similarly benefit from internal simulation. Our implementation of Sokoban procedurally generates a new level each episode (see Appendix D.4 for details, Fig. 3 for examples). This means an agent cannot memorize specific puzzles.[3] Together with the planning aspect, this makes for a very challenging environment for our model-free baseline agents, which solve less than 60% of the levels after a billion steps of training (details below). We provide videos of agents playing our version of Sokoban online [24].

While the underlying game logic operates in a $10 \times 10$ grid world, our agents were trained directly on RGB sprite graphics as shown in Fig. 4 (image size $80 \times 80$ pixels). There are no aspects of I2As that make them specific to grid world games.

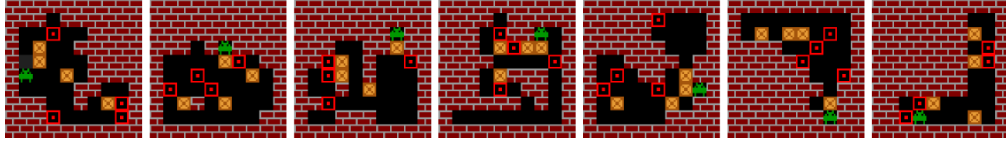

Figure 3: *Random examples of procedurally generated Sokoban levels*. The player (green sprite) needs to push all 4 boxes onto the red target squares to solve a level, while avoiding irreversible mistakes. Our agents receive sprite graphics (shown above) as observations.

## 4.1 I2A performance vs. baselines on Sokoban

Figure 4 (left) shows the learning curves of the I2A architecture and various baselines explained throughout this section. First, we compare I2A (with rollouts of length 5) against the standard model-free agent. I2A clearly outperforms the latter, reaching a performance of 85% of levels solved vs. a maximum of under 60% for the baseline. The baseline with increased capacity reaches 70% - still significantly below I2A. Similarly, for Sokoban, I2A far outperforms the copy-model.

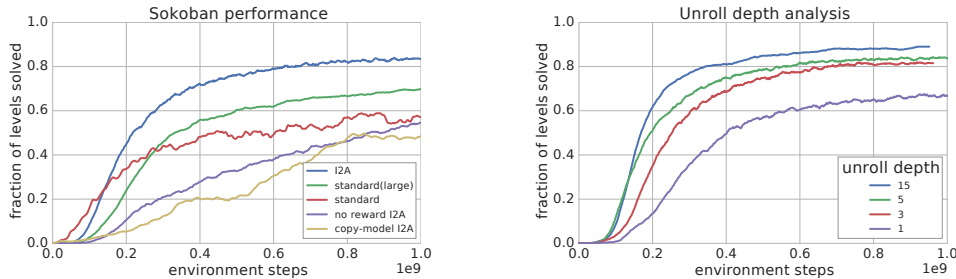

Figure 4: *Sokoban learning curves. Left:* training curves of I2A and baselines. Note that I2A use additional environment observations to pretrain the environment model, see main text for discussion. *Right:* I2A training curves for various values of imagination depth.

Since using imagined rollouts is helpful for this task, we investigate how the length of individual rollouts affects performance. The latter was one of the hyperparameters we searched over. A breakdown by number of unrolling/imagination steps in Fig. 4 (right) shows that using longer rollouts, while not increasing the number of parameters, increases performance: 3 unrolling steps improves speed of learning and top performance significantly over 1 unrolling step, 5 outperforms 3, and as a test for significantly longer rollouts, 15 outperforms 5, reaching above 90% of levels solved. However, in general we found diminishing returns with using I2A with longer rollouts. It is noteworthy that 5 steps is relatively small compared to the number of steps taken to solve a level, for which our best agents need about 50 steps on average. This implies that even such short rollouts can be highly informative. For example, they allow the agent to learn about moves it cannot recover from (such as pushing boxes against walls, in certain contexts). Because I2A with rollouts of length 15 are significantly slower, in the rest of this section, we choose rollouts of length 5 to be our canonical I2A architecture.

It terms of data efficiency, it should be noted that the environment model in the I2A was pretrained (see Section 3.2). We conservatively measured the total number of frames needed for pretraining to be lower than 1e8. Thus, even taking pretraining into account, I2A outperforms the baselines after seeing about 3e8 frames in total (compare again Fig. 4 (left)). Of course, data efficiency is even better if the environment model can be reused to solve multiple tasks in the same environment (Section 5).

## 4.2 Learning with imperfect models

One of the key strengths of I2As is being able to handle learned and thus potentially imperfect environment models. However, for the Sokoban task, our learned environment models actually perform quite well when rolling out imagined trajectories. To demonstrate that I2As can deal with less reliable predictions, we ran another experiment where the I2A used an environment model that had shown much worse performance (due to a smaller number of parameters), with strong artifacts accumulating over iterated rollout predictions (Fig. 5, left). As Fig. 5 (right) shows, even with such a

clearly flawed environment model, I2A performs similarly well. This implies that I2As can learn to ignore the latter parts of the rollout as errors accumulate, but still use initial predictions when errors are less severe. Finally, note that in our experiments, surprisingly, the I2A agent with poor model ended outperforming the I2A agent with good model. We posit this was due to random initialization, though we cannot exclude the noisy model providing some form of regularization — more work will be required to investigate this effect.

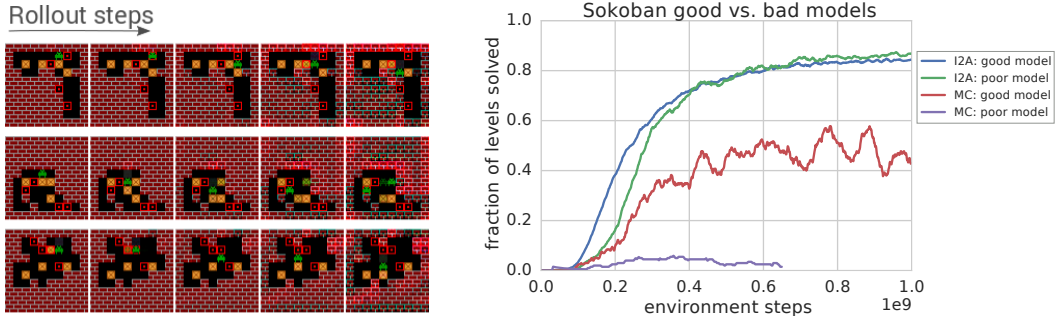

Figure 5: *Experiments with a noisy environment model. Left:* each row shows an example 5-step rollout after conditioning on an environment observation. Errors accumulate and lead to various artefacts, including missing or duplicate sprites. *Right:* comparison of Monte-Carlo (MC) search and I2A when using either the accurate or the noisy model for rollouts.

*Learning* a rollout encoder is what enables I2As to deal with imperfect model predictions. We can further demonstrate this point by comparing them to a setup without a rollout encoder: as in the classic Monte-Carlo search algorithm of Tesauro and Galperin [25], we now explicitly estimate the value of each action from rollouts, rather than learning an arbitrary encoding of the rollouts, as in I2A. We then select actions according to those values. Specifically, we learn a value function $V$ from states, and, using a rollout policy $\hat{\pi}$, sample a trajectory rollout for each initial action, and compute the corresponding estimated Monte Carlo return $\sum_{t \leq T} \gamma^t r_t^a + V(x_T^a)$ where $((x_t^a, r_t^a))_{t=0..T}$ comes from a trajectory initialized with action a. Action $a$ is chosen with probability proportional to $\exp(-(\sum_{t=0..T} \gamma^t r_t^a + V(x_T^a))/\delta)$, where $\delta$ is a learned temperature. This can be thought of as a form of I2A with a fixed summarizer (which computes returns), no model-free path, and very simple policy head. In this architecture, only $V, \hat{\pi}$ and $\delta$ are learned.[4]

We ran this rollout encoder-free agent on Sokoban with both the accurate and the noisy environment model. We chose the length of the rollout to be optimal for each environment model (from the same range as for I2A, i.e. from 1 to 5). As can be seen in Fig. 5 (right),[5] when using the high accuracy environment model, the performance of the encoder-free agent is similar to that of the baseline standard agent. However, unlike I2A, its performance degrades catastrophically when using the poor model, showcasing the susceptibility to model misspecification.

## 4.3 Further insights into the workings of the I2A architecture

So far, we have studied the role of the rollout encoder. To show the importance of various other components of the I2A, we performed additional control experiments. Results are plotted in Fig. 4 (left) for comparison. First, I2A with the copy model (Section 3.3) performs far worse, demonstrating that the environment model is indeed crucial. Second, we trained an I2A where the environment model was predicting no rewards, only observations. This also performed worse. However, after much longer training (3e9 steps), these agents did recover performance close to that of the original I2A (see Appendix D.2), which was never the case for the baseline agent even with that many steps. Hence, reward prediction is helpful but not absolutely necessary in this task, and imagined observations alone are informative enough to obtain high performance on Sokoban. Note this is in contrast to many classical planning and model-based reinforcement learning methods, which often rely on reward prediction.

### 4.4 Imagination efficiency and comparison with perfect-model planning methods

| | |
|---|---|
| I2A @87 | $\sim 1400$ |
| I2A MC search @95 | $\sim 4000$ |
| MCTS @87 | $\sim 25000$ |
| MCTS @95 | $\sim 100000$ |
| Random search | $\sim$ millions |

Table 1: Imagination efficiency of various architectures.

| Boxes | 1 | 2 | 3 | 4 | 5 | 6 | 7 |
|---|---|---|---|---|---|---|---|
| I2A (%) | 99.5 | 97 | 92 | 87 | 77 | 66 | 53 |
| Standard (%) | 97 | 87 | 72 | 60 | 47 | 32 | 23 |

Table 2: Generalization of I2A to environments with different number of boxes.

In previous sections, we illustrated that I2As can be used to efficiently solve planning problems and can be robust in the face of model misspecification. Here, we ask a different question – if we *do* assume a nearly perfect model, how does I2A compare to competitive planning methods? Beyond raw performance we focus particularly on the efficiency of planning, i.e. the number of imagination steps required to solve a fixed ratio of levels. We compare our regular I2A agent to a variant of Monte Carlo Tree Search (MCTS), which is a modern guided tree search algorithm [12, 26]. For our MCTS implementation, we aimed to have a strong baseline by using recent ideas: we include transposition tables [27], and evaluate the returns of leaf nodes by using a value network (in this case, a deep residual value network trained with the same total amount of data as I2A; see appendix D.3 for further details).

Running MCTS on Sokoban, we find that it can achieve high performance, but at a cost of a much higher number of necessary environment model simulation steps: MCTS reaches the I2A performance of $87\%$ of levels solved when using 25k model simulation steps on average to solve a level, compared to 1.4k environment model calls for I2A. Using even more simulation steps, MCTS performance increases further, e.g. reaching $95\%$ with 100k steps.

If we assume access to a high-accuracy environment model (including the reward prediction), we can also push I2A performance further, by performing basic Monte-Carlo search with a trained I2A for the rollout policy: we let the agent play whole episodes in simulation (where I2A itself uses the environment model for short-term rollouts, hence corresponding to using a model-within-a-model), and execute a successful action sequence if found, up to a maximum number of retries; this is reminiscent of nested rollouts [28]. With a fixed maximum of 10 retries, we obtain a score of $95\%$ (up from $87\%$ for the I2A itself). The total average number of model simulation steps needed to solve a level, including running the model in the outer loop, is now 4k, again much lower than the corresponding MCTS run with 100k steps. Note again, this approach requires a nearly perfect model; we don't expect I2A with MC search to perform well with approximate models. See Table 1 for a summary of the imagination efficiency for the different methods.

### 4.5 Generalization experiments

Lastly, we probe the generalization capabilities of I2As, beyond handling random level layouts in Sokoban. Our agents were trained on levels with 4 boxes. Table 2 shows the performance of I2A when such an agent was tested on levels with different numbers of boxes, and that of the standard model-free agent for comparison. We found that I2As generalizes well; at 7 boxes, the I2A agent is still able to solve more than half of the levels, nearly as many as the standard agent on 4 boxes.

## 5 Learning one model for many tasks in MiniPacman

In our final set of experiments, we demonstrate how a single model, which provides the I2A with a general understanding of the dynamics governing an environment, can be used to solve a collection of different tasks. We designed a simple, light-weight domain called MiniPacman, which allows us to easily define multiple tasks in an environment with shared state transitions and which enables us to do rapid experimentation.

In MiniPacman (Fig. 6, left), the player explores a maze that contains food while being chased by ghosts. The maze also contains power pills; when eaten, for a fixed number of steps, the player moves faster, and the ghosts run away and can be eaten. These dynamics are common to all tasks. Each task

is defined by a vector $w_{\text{rew}} \in \mathbb{R}^5$, associating a reward to each of the following five events: moving, eating food, eating a power pill, eating a ghost, and being eaten by a ghost. We consider five different reward vectors inducing five different tasks. Empirically we found that the reward schemes were sufficiently different to lead to very different high-performing policies[6] (for more details on the game and tasks, see appendix C.

To illustrate the benefits of model-based methods in this multi-task setting, we train a single environment model to predict both observations (frames) and events (as defined above, e.g. "eating a ghost"). Note that the environment model is effectively shared across all tasks, so that the marginal cost of learning the model is nil. During training and testing, the I2As have access to the frame and reward predictions generated by the model; the latter was computed from model event predictions and the task reward vector $w_{\text{rew}}$. As such, the reward vector $w_{\text{rew}}$ can be interpreted as an 'instruction' about which task to solve in the same environment [cf. the Frostbite challenge of 11]. For a fair comparison, we also provide all baseline agents with the event variable as input.[7]

We trained baseline agents and I2As separately on each task. Results in Fig. 6 (right) indicate the benefit of the I2A architecture, outperforming the standard agent in all tasks, and the copy-model baseline in all but one task. Moreover, we found that the performance gap between I2As and baselines is particularly high for tasks 4 & 5, where rewards are particularly sparse, and where the anticipation of ghost dynamics is especially important. We posit that the I2A agent can leverage its environment and reward model to explore the environment much more effectively.

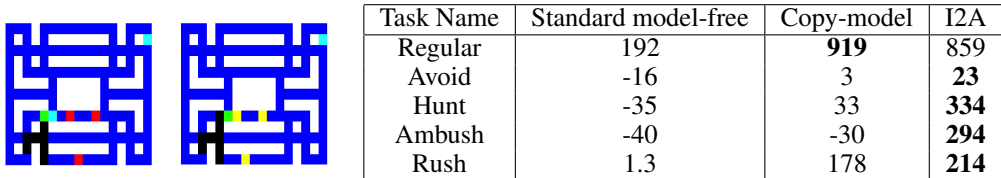

| Task Name | Standard model-free | Copy-model | I2A |
|---|---|---|---|
| Regular | 192 | **919** | 859 |
| Avoid | -16 | 3 | **23** |
| Hunt | -35 | 33 | **334** |
| Ambush | -40 | -30 | **294** |
| Rush | 1.3 | 178 | **214** |

Figure 6: *Minipacman environment. Left:* Two frames from a minipacman game. Frames are $15 \times 19$ RGB images. The player is green, dangerous ghosts red, food dark blue, empty corridors black, power pills in cyan. After eating a power pill (right frame), the player can eat the 4 weak ghosts (yellow). *Right:* Performance after 300 million environment steps for different agents and all tasks. Note I2A clearly outperforms the other two agents on all tasks with sparse rewards.

## 6 Related work

Some recent work has focused on applying deep learning to model-based RL. A common approach is to learn a neural model of the environment, including from raw observations, and use it in classical planning algorithms such as trajectory optimization [29–31]. These studies however do not address a possible mismatch between the learned model and the true environment.

Model imperfection has attracted particular attention in robotics, when transferring policies from simulation to real environments [32–34]. There, the environment model is given, not learned, and used for pretraining, not planning at test time. Liu et al. [35] also learn to extract information from trajectories, but in the context of imitation learning. Bansal et al. [36] take a Bayesian approach to model imperfection, by selecting environment models on the basis of their actual control performance.

The problem of making use of imperfect models was also approached in simplified environment in Talvitie [18, 19] by using techniques similar to scheduled sampling [37]; however these techniques break down in stochastic environments; they mostly address the compounding error issue but do not address fundamental model imperfections.

A principled way to deal with imperfect models is to capture model uncertainty, e.g. by using Gaussian Process models of the environment, see Deisenroth and Rasmussen [15]. The disadvantage of this method is its high computational cost; it also assumes that the model uncertainty is well calibrated and lacks a mechanism that can learn to compensate for possible miscalibration of uncertainty. Cutler et al. [38] consider RL with a hierarchy of models of increasing (known) fidelity. A recent multi-task

GP extension of this study can further help to mitigate the impact of model misspecification, but again suffers from high computational burden in large domains, see Marco et al. [39].

A number of approaches use models to create additional synthetic training data, starting from Dyna [40], to more recent work e.g. Gu et al. [41] and Venkatraman et al. [42]; these models increase data efficiency, but are not used by the agent at test time.

Tamar et al. [43], Silver et al. [44], and Oh et al. [45] all present neural networks whose architectures mimic classical iterative planning algorithms, and which are trained by reinforcement learning or to predict user-defined, high-level features; in these, there is no explicit environment model. In our case, we use explicit environment models that are trained to predict low-level observations, which allows us to exploit additional unsupervised learning signals for training. This procedure is expected to be beneficial in environments with sparse rewards, where unsupervised modelling losses can complement return maximization as learning target as recently explored in Jaderberg et al. [46] and Mirowski et al. [47].

Internal models can also be used to improve the credit assignment problem in reinforcement learning: Henaff et al. [48] learn models of discrete actions environments, and exploit the effective differentiability of the model with respect to the actions by applying continuous control planning algorithms to derive a plan; Schmidhuber [49] uses an environment model to turn environment cost minimization into a network activity minimization.

Kansky et al. [50] learn symbolic networks models of the environment and use them for planning, but are given the relevant abstractions from a hand-crafted vision system.

Close to our work is a study by Hamrick et al. [51]: they present a neural architecture that queries learned expert models, but focus on meta-control for continuous contextual bandit problems. Pascanu et al. [52] extend this work by focusing on explicit planning in sequential environments, and learn how to construct a plan iteratively.

The general idea of learning to leverage an internal model in arbitrary ways was also discussed by Schmidhuber [53].

# 7 Discussion

We presented I2A, an approach combining model-free and model-based ideas to implement *imagination-augmented RL*: learning to interpret environment models to augment model-free decisions. I2A outperforms model-free baselines on MiniPacman and on the challenging, combinatorial domain of Sokoban. We demonstrated that, unlike classical model-based RL and planning methods, I2A is able to successfully use imperfect models (including models without reward predictions), hence significantly broadening the applicability of model-based RL concepts and ideas.

As all model-based RL methods, I2As trade-off environment interactions for computation by pondering before acting. This is essential in irreversible domains, where actions can have catastrophic outcomes, such as in Sokoban. In our experiments, the I2A was always less than an order of magnitude slower per interaction than the model-free baselines. The amount of computation can be varied (it grows linearly with the number and depth of rollouts); we therefore expect I2As to greatly benefit from advances on dynamic compute resource allocation (e.g. Graves [54]). Another avenue for future research is on abstract environment models: learning predictive models at the "right" level of complexity and that can be evaluated efficiently at test time will help to scale I2As to richer domains.

Remarkably, on Sokoban I2As compare favourably to a strong planning baseline (MCTS) with a perfect environment model: at comparable performance, I2As require far fewer function calls to the model than MCTS, because their model rollouts are guided towards relevant parts of the state space by a learned rollout policy. This points to further potential improvement by training rollout policies that "learn to query" imperfect models in a task-relevant way.

**Acknowledgements**

We thank Victor Valdes for designing and implementing the Sokoban environment, Joseph Modayil for reviewing an early version of this paper, and Ali Eslami, Hado Van Hasselt, Neil Rabinowitz, Tom Schaul, Yori Zwols for various help and feedback.

## Footnotes

[2]The choice of forward, backward or bi-directional processing seems to have relatively little impact on the performance of the I2A, however, and should not preclude investigating different strategies.

[3]Out of 40 million levels generated, less than 0.7% were repeated. Training an agent on 1 billion frames requires less than 20 million episodes.

[4]the rollout policy is still learned by distillation from the output policy

[5]Note: the MC curves in Fig. 5 only used a single agent rather than averages.

[6]For example, in the 'avoid' game, any event is negatively rewarded, and the optimal strategy is for the agent to clear a small space from food and use it to continuously escape the ghosts.

[7]It is not necessary to provide the reward vector $w_{\text{rew}}$ to the baseline agents, as it is equivalent a constant bias.

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
