[Supplementary Material · I2A_camera_supplement.pdf]

# Supplementary material for:
# Imagination-Augmented Agents
# for Deep Reinforcement Learning

## A   Training and rollout policy distillation details

Each agent used in the paper defines a stochastic policy, i.e. a categorical distribution $\pi(a_t|o_t;\theta)$ over discrete actions $a$. The logits of $\pi(a_t|o_t;\theta)$ are computed by a neural network with parameters $\theta$, taking observation $o_t$ at timestep $t$ as input. During training, to increase the probability of rewarding actions being taken, A3C applies an update $\Delta\theta$ to the parameters $\theta$ using policy gradient $g(\theta)$:

$$g(\theta) = \nabla_\theta \log\pi(a_t|o_t;\theta)A(o_t,a_t)$$

where $A(o_t,a_t)$ is an estimate of the advantage function [55]. In practice, we learn a value function $V(o_t;\theta_v)$ and use it to compute the advantage as the difference of the bootstrapped $k$-step return and and the current value estimate:

$$A(o_t,a_t) = \left(\sum_{t\le t'\le t+k}\gamma^{t'-t}r_{t'}\right) + \gamma^{k+1}V(o_{t+k+1};\theta_v) - V(o_t;\theta_v).$$

The value function $V(o_t;\theta_v)$ is also computed as the output of a neural network with parameters $\theta_v$. The input to the value function network was chosen to be the second to last layer of the policy network that computes $\pi$. The parameter $\theta_v$ are updated with $\Delta\theta_v$ towards bootstrapped $k$-step return:

$$g(\theta_v) = -A(o_t,a_t)\partial_{\theta_v}V(o_t;\theta_v)$$

In our numerical implementation, we express the above updates as gradients of a corresponding surrogate loss [56].   To this surrogate loss, we add an entropy regularizer of $\lambda_{\text{ent}}\sum_{a_t}\pi(a_t|o_t;\theta)\log\pi(a_t|o_t;\theta)$ to encourage exploration, with $\lambda_{\text{ent}} = 10^{-2}$ throughout all experiments. Where applicable, we add a loss for policy distillation consisting of the cross-entropy between $\pi$ and $\hat\pi$:

$$l_{\text{dist}}(\pi,\hat\pi)(o_t) = \lambda_{\text{dist}}\sum_a \pi(a|o_t)\log\hat\pi(a|o_t),$$

with scaling parameter $\lambda_{\text{dist}}$. Here $\bar\pi$ denotes that we do not backpropagate gradients of $l_{\text{dist}}$ wrt. to the parameters of the rollout policy through the behavioral policy $\pi$. Finally, even though we pre-trained our environment models, in principle we can also learn it jointly with the I2A agent by a adding an appropriate log-likelihood term of observations under the model. We will investigate this in future research. We optimize hyperparameters (learning rate and momentum of the RMSprop optimizer, gradient clipping parameter, distillation loss scaling $\lambda_{\text{dist}}$ where applicable) separately for each agent (I2A and baselines).

## B   Agent and model architecture details

We used rectified linear units (ReLUs) between all hidden layers of all our agents. For the environment models, we used leaky ReLUs with a slope of $0.01$.

### B.1   Agents

#### Standard model-free baseline agent

The standard model-free baseline agent, taken from [3], is a multi-layer convolutional neural network (CNN), taking the current observation $o_t$ as input, followed by a fully connected (FC) hidden layer.

This FC layer feeds into two heads: into a FC layer with one output per action computing the policy logits $\log \pi(a_t|o_t, \theta)$; and into another FC layer with a single output that computes the value function $V(o_t; \theta_v)$. The sizes of the layers were chosen as follows:

- for MiniPacman: the CNN has two layers, both with 3x3 kernels, 16 output channels and strides 1 and 2; the following FC layer has 256 units

- for Sokoban: the CNN has three layers with kernel sizes 8x8, 4x4, 3x3, strides of 4, 2, 1 and number of output channels 32, 64, 64; the following FC has 512 units

**I2A**

The model free path of the I2A consists of a CNN identical to one of the standard model-free baseline (without the FC layers). The rollout encoder processes each frame generated by the environment model with another identically sized CNN. The output of this CNN is then concatenated with the reward prediction (single scalar broadcast into frame shape). This feature is the input to an LSTM with 512 (for Sokoban) or 256 (for MiniPacman) units. The same LSTM is used to process all 5 rollouts (one per action); the last output of the LSTM for all rollouts are concatenated into a single vector $c_{\mathrm{ia}}$ of length 2560 for Sokoban, and 1280 on MiniPacman. This vector is concatenated with the output $c_{\mathrm{mf}}$ of the model-free CNN path and is fed into the fully connected layers computing policy logits and value function as in the baseline agent described above.

**Copy-model**

The copy-model agent has the exact same architecture as the I2A, with the exception of the environment model being replaced by the identity function (constantly returns the input observation).

## B.2 Environment models

For the I2A, we pre-train separate auto-regressive models of order 1 for the raw pixel observations of the MiniPacman and Sokoban environments (see figures 7 and 8) . In both cases, the input to the model consisted of the last observation $o_t$, and a broadcasted, one-hot representation of the last action $a_t$. Following previous studies, the outputs of the models were trained to predict the next frame $o_{t+1}$ by stochastic gradient decent on the Bernoulli cross-entropy between network outputs and data $o_{t+1}$.

The Sokoban model is a simplified case of the MiniPacman model; the Sokoban model is nearly entirely local (save for the reward model), while the MiniPacman model needs to deal with nonlocal interaction (movement of ghosts is affected by position of Pacman, which can be arbitrarily far from the ghosts).

**MiniPacman model**

The input and output frames were of size 15 x 19 x 3 (width x height x RGB). The model is depicted in figure 7. It consisted of a size preserving, multi-scale CNN architecture with additional fully connected layers for reward prediction. In order to capture long-range dependencies across pixels, we also make use of a layer we call *pool-and-inject*, which applies global max-pooling over each feature map and broadcasts the resulting values as feature maps of the same size and concatenates the result to the input. Pool-and-inject layers are therefore size-preserving layers which communicate the max-value of each layer globally to the next convolutional layer.

**Sokoban model**

The Sokoban model was chosen to be a residual CNN with an additional CNN / fully-connected MLP pathway for predicting rewards. The input of size 80x80x3 was first processed with convolutions with a large 8x8 kernel and stride of 8. This reduced representation was further processed with two size preserving CNN layers before outputting a predicted frame by a 8x8 convolutional layer.

Figure 7: The minipacman environment model. The overview is given in the right panel with blow-ups of the basic convolutional building block (middle panel) and the pool-and-inject layer (left panel). The basic build block has three hyperparameters $n_1, n_2, n_3$ determining the number of channels in the convolutions; their numeric values are given in the right panel.

Figure 8: The sokoban environment model.

## C  MiniPacman additional details

MiniPacman is played in a $15 \times 19$ grid-world. Characters, the ghosts and Pacman, move through a maze. Walls positions are fixed. At the start of each level 2 power pills, a number of ghosts, and Pacman are placed at random in the world. Food is found on every square of the maze. The number of ghosts on level $k$ is $1 + \frac{\text{level} - 1}{2}$ rounded down, where level $= 1$ on the first level.

**Game dynamics**

Ghosts always move by one square at each time step. Pacman usually moves by one square, except when it has eaten a power pill, which makes it move by two squares at a time. When moving by 2 squares, if Pacman new position ends up inside a wall, then it is moved back by one square to get back to a corridor.

We say that Pacman and a ghost meet when they either end up at the same location, or when their path crosses (even if they do not end up at the same location). When Pacman moves to a square with food or a power pill, it eats it. Eating a power pill gives Pacman super powers, such as moving at

double speed and being able to eat ghosts. The effects of eating a power pill last for 19 time steps. When Pacman meets a ghost, either Pacman dies eaten by the ghost, or, if Pacman has recently eaten a power pill, the ghost dies eaten by Pacman.

If Pacman has eaten a power pill, ghosts try to flee from Pacman. They otherwise try to chase Pacman. A more precise algorithm for the movement of a ghost is given below in pseudo code:

---

**Algorithm 1** move ghost

---

```
 1: function MOVEGHOST
 2:     Inputs: Ghost object                          ▷ Contains position and some helper methods
 3:     PossibleDirections ← [DOWN, LEFT, RIGHT, UP]
 4:     CurrentDirection ← Ghost.current_direction
 5:     AllowedDirections ← []
 6:     for dir in PossibleDirections do
 7:         if Ghost.can_move(dir) then
 8:             AllowedDirections + = [dir]
 9:         if len(AllowedDirections) == 2 then    ▷ We are in a straight corridor, or at a bend
10:             if Ghost.current_direction in AllowedDirections then
11:                 return Ghost.current_direction
12:             if opposite(Ghost.current_direction) == AllowedDirections[0] then
13:                 return AllowedDirections[1]
14:             return AllowedDirections[0]
15:         else                                                  ▷ We are at an intersection
16:             if opposite(Ghost.current_direction) in AllowedDirections then
17:                 AllowedDirections.remove(opposite(Ghost.current_direction))  ▷ Ghosts do
                        not turn around
18:             X = normalise(Pacman.position - Ghost.position)
19:             DotProducts = []
20:             for dir in AllowedDirections do
21:                 DotProducts + = [dot_product(X, dir)]
22:             if Pacman.ate_super_pill then
23:                 return AllowedDirections[argmin(DotProducts)]     ▷ Away from Pacman
24:             else
25:                 return AllowedDirections[argmax(DotProducts)]         ▷ Towards Pacman
```

---

**Task collection**

We used 5 different tasks available in MiniPacman. They all share the same environment dynamics (layout of maze, movement of ghosts, ... ), but vary in their reward structure and level termination. The rewards associated with various events for each tasks are given in the table below.

| Task | At each step | Eating food | Eating power pill | Eating ghost | Killed by ghost |
|------|------|------|------|------|------|
| Regular | 0 | 1 | 2 | 5 | 0 |
| Avoid | 0.1 | -0.1 | -5 | -10 | -20 |
| Hunt | 0 | 0 | 1 | 10 | -20 |
| Ambush | 0 | -0.1 | 0 | 10 | -20 |
| Rush | 0 | -0.1 | 10 | 0 | 0 |

When a level is cleared, a new level starts. Tasks also differ in the way a level was *cleared*.

- Regular: level is cleared when all the food is eaten;

- Avoid: level is cleared after 128 steps;

- Hunt: level is cleared when all ghosts are eaten or after 80 steps.

- Ambush: level is cleared when all ghosts are eaten or after 80 steps.

- Rush: level is cleared when all power pills are eaten.

Figure 9: The pink bar appears when Pacman eats a power pill, and it decreases in size over the duration of the effect of the pill.

There are no lives, and episode ends when Pacman is eaten by a ghost.

The time left before the effect of the power pill wears off is shown using a pink shrinking bar at the bottom of the screen as in Fig. 9.

**Training curves**

Figure 10: Learning curves for different agents and various tasks

# D    Sokoban additional details

## D.1    Sokoban environment

In the game of Sokoban, random actions on the levels would solve levels with vanishing probability, leading to extreme exploration issues for solving the problem with reinforcement learning. To alleviate this issue, we use a shaping reward scheme for our version of Sokoban:

- Every time step, a penalty of -0.1 is applied to the agent.
- Whenever the agent pushes a box on target, it receives a reward of +1.
- Whenever the agent pushes a box off target, it receives a penalty of -1.
- Finishing the level gives the agent a reward of +10 and the level terminates.

The first reward is to encourage agents to finish levels faster, the second to encourage agents to push boxes onto targets, the third to avoid artificial reward loop that would be induced by repeatedly pushing a box off and on target, the fourth to strongly reward solving a level. Levels are interrupted after 120 steps (i.e. agent may bootstrap from a value estimate of the last frame, but the level resets to a new one). Identical levels are nearly never encountered during training or testing (out of 40 million levels generated, less than 0.7% were repeated). Note that with this reward scheme, it is always optimal to solve the level (thus our shaping scheme is valid). An alternative strategy would have been to have the agent play through a curriculum of increasingly difficult tasks; we expect both strategies to work similarly.

## D.2 Additional experiments

Our first additional experiment compared I2A with and without reward prediction, trained over a longer horizon. I2A with reward prediction clearly converged shortly after 1e9 steps and we therefore interrupted training; however, I2A without reward prediction kept increasing performance, and after 3e9 steps, we recover a performance level of close to 80% of levels solved, see Fig. 11.

Figure 11: I2A with and without reward prediction, longer training horizon.

Next, we investigated the I2A with Monte-Carlo search (using a near perfect environment model of Sokoban). We let the agent try to solve the levels up to 16 times within its internal model. The base I2A architecture was solving around 87% of levels; mental retries boosted its performance to around 95% of levels solved. Although the agent was allowed up to 16 mental retries, in practice all the performance increase was obtained within the first 10 mental retries. Exact percentage gain by each mental retry is shown in Fig. 12. Note in Fig. 12, only 83% of the levels are solved on the first mental attempt, even though the I2A architecture could solve around 87% of levels. The gap is explained by the use of an environment model: although it looks nearly perfect to the naked eye, the model is not actually equivalent to the environment.

Figure 12: Gain in percentage by each additional mental retry using a near perfect environment model.

### D.3   Planning with the perfect model and Monte-Carlo Tree Search in Sokoban

We first trained a value network that estimates the value function of a trained model-free policy; to do this, we trained a model-free agent for 1e9 environment steps. This agent solved close to 60 % of episodes. Using this agent, we generated 1e8 (frame, return) pairs, and trained the value network to predict the value (expected return) from the frame; training and test error were comparable, and we don't expect increasing the number of training points would have significantly improved the quality of the the value network.

The value network architecture is a residual network which stacks one convolution layer and 3 convolution blocks with a final fully-connected layer of 128 hidden units. The first convolution is $1 \times 1$ convolution with 128 feature maps. Each of the three residual convolution block is composed of two convolutional layers; the first is a $1 \times 1$ convolution with 32 feature maps, the second a $3 \times 3$ convolution with 32 feature maps, and the last a $1 \times 1$ layer with 128 feature maps. To help the value networks, we trained them not on the pixel representation, but on a $10 \times 10 \times 4$ symbolic representation.

The trained value network is then employed during search to evaluate leaf-nodes — similar to [12], replacing the role of traditional random rollouts in MCTS. The tree policy uses [57, 58] with a fine-tuned exploration constant of 1. Depth-wise transposition tables for the tree nodes are used to deal with the symmetries in the Sokoban environment. External actions are selected by taking the max Q value at the root node. The tree is reused between steps but selecting the appropriate subtree as the root node for the next step.

Reported results are obtained by averaging the results over 250 episodes.

### D.4   Level Generation for Sokoban

We detail here our procedural generation for Sokoban levels - we follow closely methods described in [59, 60].

The generation of a Sokoban level involves three steps: room topology generation, position configuration and room reverse-playing. Topology generation: Given an initial width*height room entirely constituted by wall blocks, the topology generation consists in creating the 'empty' spaces (i.e. corridors) where boxes, targets and the player can be placed. For this simple random walk algorithm with a configurable number of steps is applied: a random initial position and direction are chosen. Afterwards, for every step, the position is updated and, with a probability $p = 0.35$, a new random direction is selected. Every 'visited' position is emptied together with a number of surrounding wall blocks, selected by randomly choosing one of the following patterns indicating the adjacent room blocks to be removed (the darker square represents the reference position, that is, the position being visited). Note that the room 'exterior' walls are never emptied, so from a width×height room only a (width-2)×(height-2) space can actually be converted into corridors. The random walk approach guarantees that all the positions in the room are, in principle, reachable by the player. A relatively small probability of changing the walk direction favours the generation of longer corridors, while the application of a random pattern favours slightly more convoluted spaces. Position configuration:

Once a room topology is generated, the target locations for the desired N boxes and the player initial position are randomly selected. There is the obvious prerequisite of having enough empty spaces in the room to place the targets and the player but no other constraints are imposed in this step.

Reverse playing: Once the topology and targets/player positions are generated the room is reverse-played. In this case, on each step, the player has eight possible actions to choose from: simply moving or moving+pulling from a box in each possible direction (assuming for the latter, that there is a box adjacent to the player position).

Initially the room is configured with the boxes placed over their corresponding targets. From that position a depth-first search (with a configurable maximum depth) is carried out over the space of possible moves, by 'expanding' each reached player/boxes position by iteratively applying all the possible actions (which are randomly permuted on each step). An entire tree is not explored as there are different combinations of actions leading to repeated boxes/player configurations which are skipped.

Statistics are collected for each boxes/player configuration, which is, in turn, scored with a simple heuristic:

$$\text{RoomScore} = \text{BoxSwaps} \times \sum_i \text{BoxDisplacement}_i$$

where BoxSwaps represents the number of occasions in which the player stopped pulling from a given box and started pulling from a different one, while BoxDisplacement represents the Manhattan distance between the initial and final position of a given box. Also whenever a box or the player are placed on top of one of the targets the RoomScore value is set to 0. While this scoring heuristic doesn't guarantee the complexity of the generated rooms it's aimed to a) favour room configurations where overall the boxes are further away from their original positions and b) increase the probability of a room requiring a more convoluted combination of box moves to get to a solution (by aiming for solutions with higher boxSwaps values). This scoring mechanism has empirically proved to generate levels with a balanced combination of difficulties.

The reverse playing ends when there are no more available positions to explore or when a predefined maximum number of possible room configurations is reached. The room with the higher RoomScore is then returned.

Defaul parameters:

- A maximum of 10 room topologies and for each of those 10 boxes/player positioning are retried in case a given combination doesn't produce rooms with a score > 0.
- The room configuration tree is by default limited to a maximum depth of 300 applied actions.
- The total number of visited positions is by default limited to 1000000.
- Default random-walk steps: $1.5\times$ (room width + room height).