[Reviews · NeurIPS 2017]

Reviewer 1



This paper presents an approach to model-based reinforcement learning where, instead of directly estimating the value of actions in a learned model, a neural network processes the model's predictions, combining with model-free features, to produce a policy and/or value function. The idea is that since the model is likely to be flawed, the network may be able to extract useful information from the model's predictions while ignoring unreliable information. The approach is studied in procedurally generated Sokoban puzzles and a synthetic Pac-Man-like environment and is shown to outperform purely model-free learning as well as MCTS on the learned model. ---Quality--- I feel the authors have done an exceptional job of presenting and evaluating their claims. The experiments are thorough and carefully designed to tease issues apart and to clearly answer well-stated questions about the approach. I found the experiments to provide convincing evidence that I2A is taking advantage of the learned model, is robust to model flaws, and can leverage the learned model for multiple tasks. I had one question regarding the experiments comparing I2A and MCTS. I2A was trained for millions of steps to obtain a good policy. Was a comparable amount of data used to train the MCTS leaf evaluation function? I looked in the appendix, but didn't see that detail. Also, out of curiosity, if you have access to a perfect simulator, isn't it a bit funny to make I2A plan with the learned model? Why not skip the model learning part and train I2A to interpret rollouts from the simulator? ---Clarity--- I felt the paper was exceptionally well-written. The authors did a good job of motivating the problem and their approach and clearly explained the big ideas. As is often the case with deep learning papers, it would be *extremely* difficult to reproduce these results based purely on the paper itself -- there are always lots of details regarding architecture, meta-parameters, etc. However, the appendix does a good job of narrowing this gap. I hope the authors will consider publishing source code as well. ---Originality--- As far as I am aware, this is a highly novel approach to MBRL. I have never seen anything like it, and it is fundamentally, conceptually different than existing MBRL approaches. The authors did a good job of citing relevant work (the references section is prodigious). Nevertheless, there has been some other recent work outside the context of deep nets studying the problem of planning with approximate, learned models that the authors may consider discussing (recognizing of course the limitations of space). For instance, Talvitie UAI 2014 and AAAI 2016, Jiang et al. AAMAS 2015, Venkatraman et al. ISER 2016. The presented approach is unquestionably distinct from these, but there may be connections or contrasts worth making. Also, the I2A architecture reminds me of the Dyna architecture, which similarly combines model-based and model-free updates into a single value function/policy (the difference being that it directly uses reward/transition predictions from the model). Again, I2A is definitely new, but the connection may be valuable to discuss. ---Significance--- I think this is an important paper. It is addressing an important problem in a creative and novel way, with demonstrably promising results. It seems highly likely that this paper will inspire follow-up work, since it effectively proposes a new category of MBRL algorithms that seems well-worth exploring. ***After author response*** I have read the authors' response and thank them for the clarifications.

Reviewer 2



Paper summary: This paper introduces I2A model which has both model-free and model-based components. Authors consider the setting where the exact transition model is not available and the domain is complex. The goal is to design a model that will be robust to the errors in transition function approximation. To achieve this, I2A has a rollout based on imperfect transition model and a rollout policy (which is learnt simultaneously) and encode the information from rollout and add it to the model-free information. Authors show experiments on sokobon and pacman worlds. My comments: I like the idea of using policy distillation to simultaneously learn the rollout policy. This looks like the best possible approximation of the current policy. 1. Can authors report the comparison between test performance of the I2A model and the rollout policy model? I see that rollout policy is a distilled version of the I2A model. If it performs almost close to I2A, it can be used as a cheaper substitute in time-constrained domains. 2. Authors use a pre-trained environment model and freeze it during I2A training. Even though they say that random initialization doesn’t perform well, I would like to see the performance of I2A with randomly-initialized environment model and also pretrained environmental model with fine-tuning during I2A training. These results would help us to get better understanding of the model. 3. I don’t understand the intuition behind the copy model. If the authors’ goal is to check if the number of parameters have effect in performance, they should simply increase the number of parameters in the model-free baseline and see if there is increase in performance. I think this should be one of the baselines. 4. Figure 4(right) shows the effect of increasing the rollout length. Authors claim that increasing the rollout length only increase the performance and then saturates. Can you also report what is the performance curve for rollout lengths 10 and 15? That would help us to confirm that LSTM encoding indeed helps in avoiding the multiplicative prediction errors. 5. Silver et al. [34] is an important baseline that is missing in the paper. While I understand that you use a separate transition model and they learn the transition end to end, the comparison would help us to gauge which direction is superior. I am not saying that we should not pursue research in inferior direction, but just that it would help us benchmark different approaches. It is perfectly fine if you don’t do better than Silver et al. [34]. 6. Experiment setup for section 4.2 is not clearly written. Do you remove model free path of all the models? Do you remove them only during testing or also during training? 7. Line 175-178 is either not clear or contradicting. 8. In figure 5(right) poor model finally reaches better performance. I am surprised that there is no discussion in the text explaining this. 9. Message from section 4.3 is not clear. No-reward IAA is close to baseline performance. This means that reward prediction is importance to achieve I2A performance. However authors say that reward prediction is not needed. Can you please comment on this? 10. Captions missing in table 1 and 2. 11. Your appendix is corrupted after page 4. So I cannot read from page 5. I would increase my scores if you agree to add the experiments suggested in points 2,3,4,5.

Reviewer 3



This paper introduces a novel method of combining model-based and model-free RL. The core of the architecture is a dynamics model that, when rolled out, predicts future observations and rewards. These imagined rollouts are then processed into a concise representation that is fed as additional input into a policy which is trained using model-free methods. Experiment ablations show the effect of various design choices and experimental comparisons show sample efficiency gains in complex environments compared to baseline methods. Overall I find the paper result very compelling and clearly presented. This work combines and builds upon several prior ideas (such as learning actionable imperfect models, auxiliary rewards, boosting model-free with model-based, model-based for transfer learning) into a method that works on relevant, complex tasks. The experiments provide insight into the method, which will help the research community build upon this work. I therefore vote for a strong accept. Major feedback: - Section 6: The related work is lacking. Here are some additional references you may consider, but more than what I list should be added: Dyna architecture (https://pdfs.semanticscholar.org/b5f8/a0858fb82ce0e50b55446577a70e40137aaf.pdf) Accelerating model-free with model-based (https://pdfs.semanticscholar.org/b5f8/a0858fb82ce0e50b55446577a70e40137aaf.pdf) Goal-driven dynamics learning (https://arxiv.org/pdf/1703.09260.pdf) - Supplementary B.2: The environment models being different for Sokoban and Minipacman is concerning because it indicates the environment models may be overfit to the particular environment. Therefore it is unclear how much engineering it takes to get I2A to work on a new domain, thus calling into question its practicality. Using one architecture in multiple domains (e.g., Atari suite) would address this concern. Minor feedback: - Section 2: Fig. 1 and 2 are very good - Section 3.1: The method for generating rollouts is rather unorthodox. Further investigation/insight would be helpful. - Section 4.1: Says that unrolling for > 5 did not increase performance, but would still be useful to see how much performance decreases - Section 4.2: Fig. 5 (right) is convincing - Section 4.3: If reward prediction just speeds up learning and doesn’t affect final performance, please have the learning curves show this - Section 4.4: Tables need captions - Section 4.4: When doing MCTS with I2A, does performance always increase when allowed to plan for longer? The text seems to indicate this, but if so I’m a bit surprised since the more you plan, the more likely you are to incorrectly exploit your model and get a maximization bias (motivation from double Q-learning)